# Thermo-Mechanical Modelling of Wire-Arc Additive Manufacturing (WAAM) of Semi-Finished Products

**Marcel Graf [1],\*, Andre Hälsig [2], Kevin Höfer [2], Birgit Awiszus [1] and Peter Mayr [2]** 

[1]  Institute for Machine Tools and Production Processes, Professorship Virtual Production Engineering, Chemnitz University of Technology, Reichenhainer Str. 70, 09126 Chemnitz, Germany; birgit.awiszus@mb.tu-chemnitz.de

[2]  Institute of Joining and Assembly, Professorship Welding Engineering, Chemnitz University of Technology, Reichenhainer Str. 70, 09126 Chemnitz, Germany; andre.haelsig@mb.tu-chemnitz.de (A.H.); kevin.hoefer@mb.tu-chemnitz.de (K.H.); peter.mayr@mb.tu-chemnitz.de (P.M.)

\*  Correspondence: marcel.graf@mb.tu-chemnitz.de; Tel./Fax: +49-371-531-31796

**Abstract:** Additive manufacturing processes have been investigated for some years, and are commonly used industrially in the field of plastics for small- and medium-sized series. The use of metallic deposition material has been intensively studied on the laboratory scale, but the numerical prediction is not yet state of the art. This paper examines numerical approaches for predicting temperature fields, distortions, and mechanical properties using the Finite Element (FE) software MSC Marc. For process mapping, the filler materials G4Si1 (1.5130) for steel, and AZ31 for magnesium, were first characterized in terms of thermo-physical and thermo-mechanical properties with process-relevant cast microstructure. These material parameters are necessary for a detailed thermo-mechanical coupled Finite Element Method (FEM). The focus of the investigations was on the numerical analysis of the influence of the wire feed (2.5–5.0 m/min) and the weld path orientation (unidirectional or continuous) on the temperature evolution for multi-layered walls of miscellaneous materials. For the calibration of the numerical model, the real welding experiments were carried out using the gas-metal arc-welding process—cold metal transfer (CMT) technology. A uniform wall geometry can be produced with a continuous welding path, because a more homogeneous temperature distribution results.

**Keywords:** wire arc additive manufacturing (WAAM); multi-pass welding; FEA; MSC Marc; steel G4Si1; magnesium AZ31

## 1. Introduction

Additive manufacturing processes have become very significant in recent years. They have many advantages in comparison to conventional processes. The production of complex and individual components is possible with additive manufacturing processes. In principle, the layer by layer technology can use metals to produce almost any profile. The processes generate components by the joining of meltable materials, where the geometry is continuously built [1]. Additive manufacturing processes can also be classified by EN ISO17296-2-2015 into initial material, workpiece size, or energy source [2]. In addition, metallic additive manufacturing processes are subdivided into powder-bed, powder-feed, and wire-feed systems [3,4].

Compared to the powder-bed system, wire-based additive manufacturing processes have a higher deposition rate, use cheaper filler materials, and require lower investment [5]. Additionally, the wire-arc additive manufacturing (WAAM) process, in combination with a robot, also allows the production of complex geometries [6–8]. A component manufactured by a wire-based additive manufacturing

technique can achieve the tensile strength of a forged or cast complex component [6]. However, the surface finish and accuracy is poorer than with powder-bed or beam-melting additive technologies [9]. In order to produce a high-quality product, it is important to determine the interaction of each process parameter with the component properties (microstructure, residual stresses), and to combine the process with other production technologies, if necessary [10]. An initial estimate can be made using numerical methods that take different boundary conditions into consideration. The current aim of the numerical simulation of additive manufacturing processes is the modelling of a thermo-mechanical coupled simulation without microstructural properties at mesoscopic level. Various numerical methods can be applied for this. On the one hand, the volume-of-fluid method is used for the geometrical prediction—to consider the flow behaviour of the molten wire as a drop, and to model its solidification and wetting behaviour to the nearest drop [3]. On the other hand, additive technologies, such as WAAM or wire laser metal deposition (LMD-W), can also be modelled by a Finite Element (FE) method based on the element-birth technique, for the determination of real temperature fields and residual stresses. The focus for this simulation of the WAAM process has been on calculating temperature distribution, displacement of components and residual stresses [11–16]. The focus of this paper is not on powder-bed methods.

## 2. FE Model

### 2.1. Material Data

For a transient thermo-mechanical simulation of the multi-layer welding process, it is essential that the thermo-physical properties of the filler material be determined, taking the real microstructure (dendritic microstructure) into consideration, and modelled as a function of the temperature (see Figure 1) in addition to the real chemical composition. For the investigated filler materials AZ31 (magnesium alloy) and G4Si1 (steel alloy), the necessary material data were partly taken from the literature [17] and partly calculated in the JMatPro material simulation software Version 10 (Sente Software Ltd., Guildford, United Kingdom), or identified by experiments. The material software JMatPro uses thermo-physical and thermo-dynamic correlations and calculations for the determination of material properties in the material equilibria, under consideration of the chemical composition [18]. It should be noted that the samples for the physical simulation were eroded from a welding seam, so a comparable microstructural state to the WAAM process was used in the characterisation. The physical simulation of the thermal expansion coefficient was determined in the quenching and forming dilatometer Bähr DIL 805 A/T (Bähr Thermoanalyse GmbH now TA Instruments, Hüllhorst, Germany). The specific heat capacities were measured by differential scanning calorimetry. For the characterisation of magnesium and G4Si1, the DSC-60 from Mettler Toledo (Mettler-Toledo International Inc., Columbus, OH, USA). and the Multi HTC96 from Setaram (SETARAM Instrumentation, Caluire, France). were used respectively. The experimental data represent the average of three measurements.

In the case of material data for steel, the point of phase transformation can be detected in the temperature range of $A_{C1}$ between 750 °C and 800 °C during the measurement of the specific heat capacity. Therefore, the real thermal material behaviour is regarded in the Finite Element Method (FEM).

For a holistic approach to the material behaviour in the numerical simulation, the elastic-plastic material properties are necessary. Hence, the temperature-dependent Young's modules (see Figure 2) and the flow curves (see Figure 3) for both materials under technologically relevant parameters were also established and implemented in the FE software MSC Marc (MSC Software Corporation now Hexagon AB, Newport Beach, CA, USA), as tables [14].

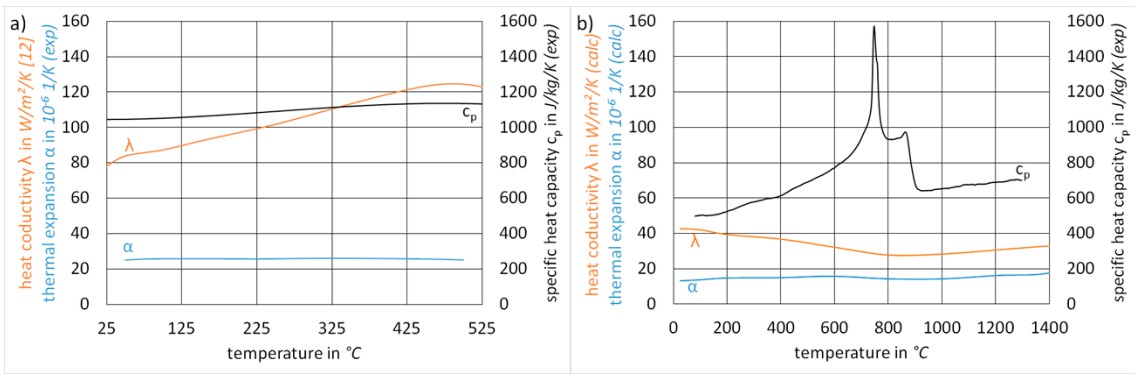

**Figure 1.** Thermo-physical properties of (**a**) magnesium AZ31 and (**b**) filler material G4Si1, taken partially from literature for magnesium [17], JMatPro for G4Si1 (calc), and partially from experiments (exp).

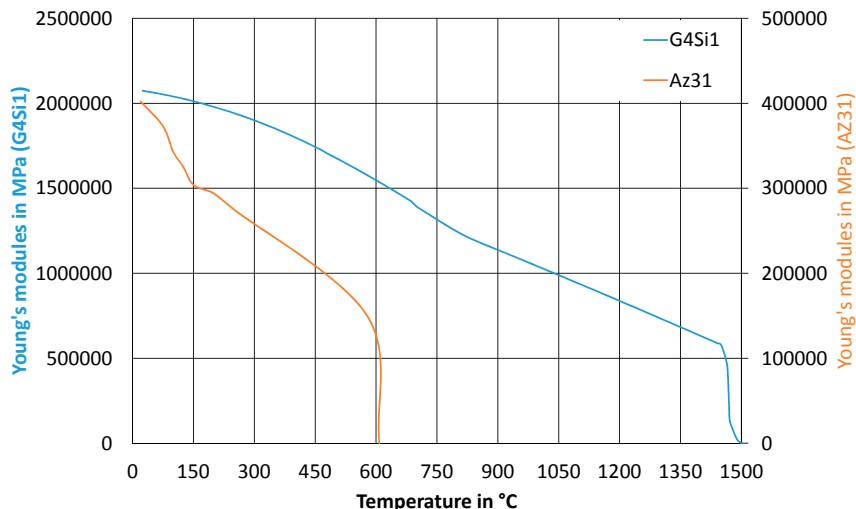

**Figure 2.** Young's modules as function of temperature for G4Si1 (blue) and AZ31 (orange).

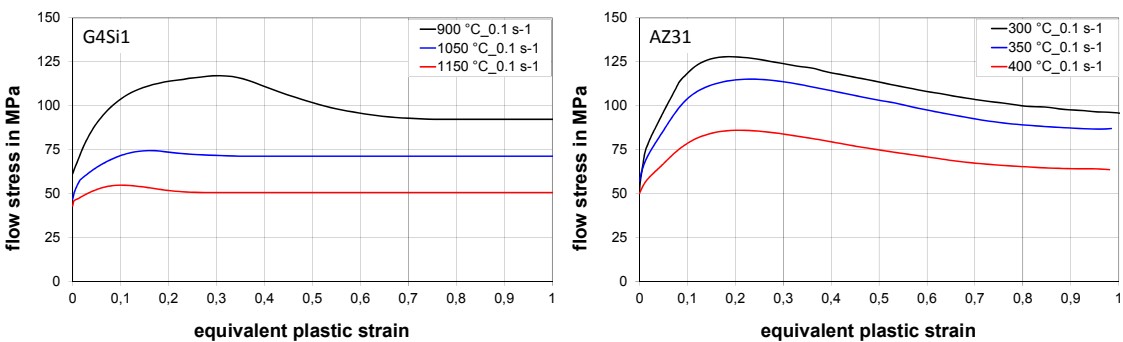

**Figure 3.** Flow curves as function of temperature for constant strain rate of 0.1 s$^{-1}$ ((**left**) G4Si1; (**right**) AZ31).

### 2.2. Boundary Conditions

Some numerical studies consider each welding seam as rectangular cross-sections [11,12,19]. In the present investigation, a realistic simplified semi-circular cross section was modelled in MSC Marc for the first layers, followed by a sickle-shape for the subsequent layers, each shifted by the resulting offset relative to the previous position. The number of the layers for the wall were the same, but the volume depended on the wire feed (Figure 4). Increasing wire feed increased the seam volume

(cross-section increase) of each layer at constant welding/deposition speed. For the additive tubes, the wire feed and welding speed were not varied.

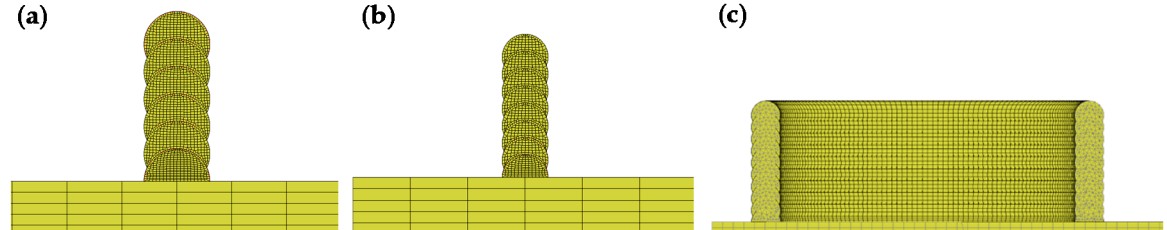

**Figure 4.** Finite Element (FE) models with the mesh: (**a**) wall with wire feed of 5 m/min; (**b**) wall with wire feed 2.5 m/min; (**c**) tube with wire feed 5 m/min.

Thus, the re-melting of the previous layer can be regarded to have a volume of 40–50% [20]. However, the introduced power/heat remained constant, so the temperature field and the thermally resulting component properties could be simulated. The geometry or volume used for each layer mainly depended on the wire diameter, wire feed and welding speed. All welding layers had gluing contacts between them, for the purposes of observing the heat transfer between them. In order that the temperature distribution could be calculated properly, the base plate was also modelled and, on the lower surface, a heat-transfer coefficient to the welding table was defined by the Thermal Face Flux condition (for steel with $\alpha_{G4Si1}$ = 2000 W/m$^2$/K, and for magnesium with $\alpha_{AZ31}$ = 100 W/m$^2$/K) [14,21]. The heat-transfer coefficient to the environment was $\alpha_{air}$ = 35 W/m$^2$/K for all numerical investigations.

In the numerical model, the double-ellipsoidal heat source according to Goldak [22] was considered. This is a typical approach for common arc-welding simulations, and replicates the thermal energy input well [13,14,19,22]. The underlying energy density function subdivides the weld pool along the welding path (*z*-axis) into the front ($c_f$) and rear areas ($c_r$), and is primarily dominated by the welding power [22].

For the simulation, the geometrical dimension of the melt pool was necessary for modelling the generated heat flow (see Table 1), which was influenced by the material-specific adjustments (process efficiency, current and voltage).

**Table 1.** Geometric parameters for describing the melt pool.

| Parameter | Symbol | Unit | G4Si1 | | AZ31 |
|---|---|---|---|---|---|
| Wire feed | $v_{wire}$ | m/min | 2.5 | 5 | 5 |
| Welding speed | $v_{welding}$ | cm/min | 40 | 40 | 40 |
| Half width | $a$ | mm | 2 | 3 | 3 |
| Depth | $b$ | mm | 2 | 3 | 3 |
| Front length | $c_f$ | mm | 2 | 2 | 2 |
| Rear length | $c_r$ | mm | 3 | 3 | 6 |

The 'element birth and death' method was applied to the investigated deposition process, for which the melting temperature was employed as the activation criterion. A melting point of $\vartheta_{melt,AZ31}$ = 630 °C [17] was defined for the AZ31 alloy, and JMatPro calculated a melting point of $\vartheta_{melt,G4Si1}$ = 1503 °C for G4Si1 based on chemical composition [14]. This modelling method deactivates all elements of the weld seam until the melting point is reached, at which point all elements are activated and their material properties are considered. If an element is activated, it remains activated up to the defined criteria—here the melting point.

As in reality, the clamping devices were also taken into account within the FEM by defining a fixed displacement at the appropriate position to calculate the residual stresses, and therefore the distortion, of the final semi-finished products. All components were meshed with the eight-noded, isoparametric,

three-dimensional brick element type 7 (8-layer wall model with around 150,000 elements, and 46-layer pipe model with around 220,000 elements). For the numerical analyses of the thermal behaviour of the semi-finished products, the number of elements was reduced in some simulations to reduce computation time (the 46-layer tube required 82 h on a 12-core processor). The base plate was meshed very coarsely compared to the seam.

## 3. Experimental Setup

A six-axis robotic handling system was used in combination with short-arc controlled gas–metal arc-welding (GMAW) process. Due to the low heat input combined with the highest possible amount of filler material per time, the cold metal transfer (CMT) welding process from Fronius International GmbH (Wels, Austria) was used as the WAAM process. Regarding the difference in base material, two types of solid filler material were used:

- G4Si1 (1.5130) steel with a diameter of 1.2 mm; shielding gas 15 L/min (82% Ar and 18% $CO_2$)
- AZ31 magnesium alloy with a diameter of 1.2 mm; shielding gas 15 L/min pure argon.

During the investigation, the welding speed was constant ($v_{welding}$ = 40 cm/min) throughout the experiments, but the wire feed $v_{wire}$ varied between 2.5 m/min and 5.0 m/min. The offset per layer was set to 1.7 mm, with no intermediate time between the single layers for cooling or to reach a defined layer temperature. During the discontinuous wall welding, there was a break of approximately 2 s for the robot to move to the start position. Various samples were thus produced (see Figure 5):

- 200-mm-long walls with minimum of 20 layers, produced by a continuous process from both sides or by discontinuous welding from one side.
- Pipes with 20 layers and a diameter of 60 mm, produced by continuous process by a combination of circles, where the *z*-level shift was done immediately after completing the circle below.

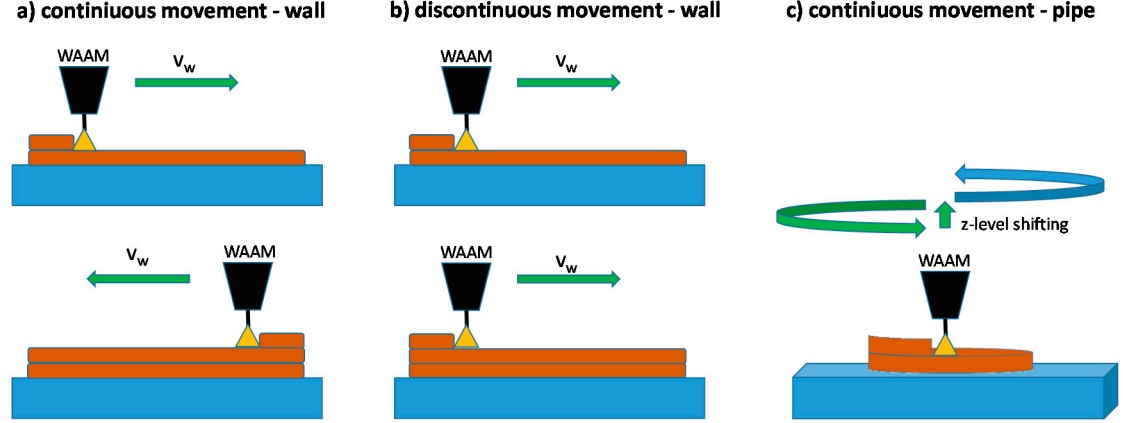

**Figure 5.** Scheme of the handling processes with continuous or discontinuous movement (left side: wall, right side: pipe).

The welding parameters measured were the average current and voltage, which depended on the filler material and the wire feed speed:

- G4Si1 with $v_{wire}$ = 2.5 m/min and current *I* = 90–110 *A*, voltage *U* = 11.5–13 *V*;
- G4Si1 with $v_{wire}$ = 5.0 m/min and current *I* = 135–165 *A*, voltage *U* = 13–14.5 *V*;
- AZ31 with $v_{wire}$ = 2.5 m/min and current *I* = 35–40 *A*, voltage *U* = 9–11 *V*;
- AZ31 with $v_{wire}$ = 5.0 m/min and current *I* = 45–55 *A*, voltage *U* = 11–12 *V*;

To analyse the temperature level, several type-K thermocouples were integrated into the sample. The drilled thermocouples were pushed into the weld pool immediately after the welding

arc. The recording time of the thermocouples started after the solidification of the filler material. This allowed the temperature–time behaviour to be analysed for defined layers and positions in the semi-finished product during welding, cooling, and reheating by the following layers.

## 4. Simulation and Experiments for WAAM

The investigations of the influence of weld path orientation and wire feed were carried out for G4Si1 and AZ31 by using a constant welding speed. Due to the different materials, different currents and voltages resulted. Figure 6 shows selected examples. In contrast to other studies in the literature, these investigations represent fast additive manufacturing, because during the layer-by-layer welding there was no break time to achieve a specific intermediate layer temperature.

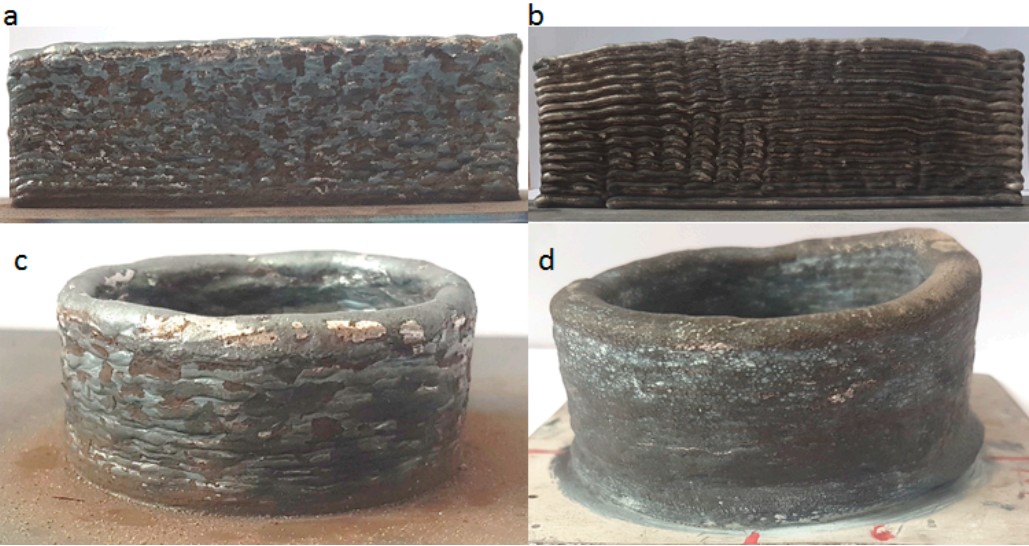

**Figure 6.** Semi-finished products, (**a**,**b**) wall and (**c**,**d**) pipe, made by WAAM with different materials, (**a**,**c**) G4Si1 and (**b**,**d**) AZ31.

- **G4Si1 Results**

The basis for the calibration of the simulation models used was provided by real measured temperature profiles during welding of G4Si1. In order to be able to validate the FE model, thermocouples were placed not only in the weld seam (approx. 50 mm after the beginning or 50 mm before the end of the seam), but also in the respective base plates, so the defined heat-transfer coefficient to the welding table could be verified. In the middle of the 5 mm thick base plates, 25 mm after the beginning of the seam (position 1), 100 mm after the beginning of the seam (=middle) and 25 mm before the seam end (position 3), holes with a diameter of 1.6 mm were drilled up to half of the width of the base plate, to measure directly in/under the heat-affected zone. The measured and simulated temperature profiles for G4Si1 are shown in Figure 7 (FEM used a 6-layer wall model; the experiment used a 15-layer wall). It was not always possible to insert the thermocouples at the same position for the measurement, because it was done manually. In any case, this had no negative impact on the calibration of the FEM.

The aim was to investigate the influence of the handling or movement technology numerically using the calibrated FE model (see Figure 8).

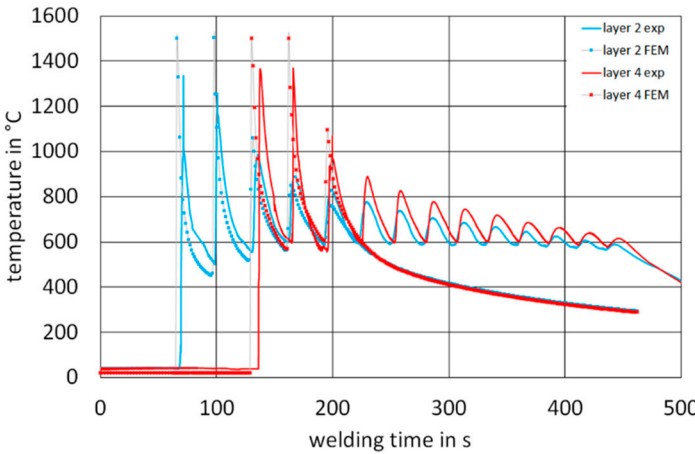

**Figure 7.** Comparison of measured (solid line) and simulated (dotted line) temperatures of the discontinuous welding process for G4Si1 with $v_{wire}$ = 5.0 m/min and $v_{welding}$ = 40 cm/min.

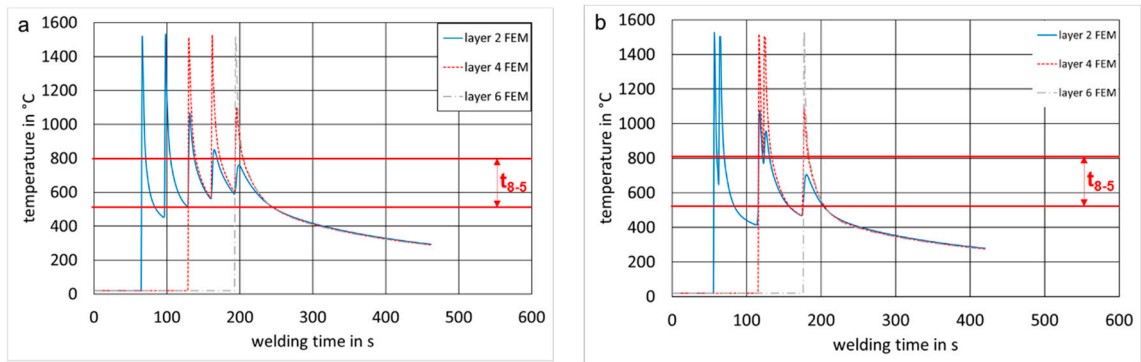

**Figure 8.** Comparison of measured and simulated temperatures depending on welding orientation for G4Si1 with $v_{wire}$ = 5.0 m/min and $v_{welding}$ = 40 cm/min: (**a**) discontinuous/unidirectional; (**b**) continuous/multidirectional.

The continuous process was significantly faster, because there was no need to move back to the subsequent starting position after finishing the welding of each layer. The comparison between simulation and experiment shows—particularly in the tests with steel filler material—that the maximum temperature cannot really be identified, because the thermocouples were not designed for the melting point of the steel, and the inertia of the measuring sensors meant they did not even record the short-term maximum temperature. However, the thermocouples were very important for the description of the cooling and reheating behaviour of each layer. Thereby, the thermal behaviour could be detected and compared with the simulated results. The analysis of the temperatures in the base plate shows that the thermal cycle oscillates more at the lower layer level. During the processes, the heating and cooling lay on a near steady state level, because the heat-affected zone was moved to higher layers and an increasing heat flow took place via the outside of the wall.

The quality of the FE results compared to the real tests was good in all cases. One result is that the welding orientation has an influence on the temperature profile and the wall geometry. During continuous multi-layered welding, the beginning and the end—depending on position—are exposed to the heat input twice within a short time and this input is symmetrical in both sides. In contrast, the welding speed and local heat input in discontinuous welding is the same in each layer, but the weld is not symmetrical. The base layer is colder at the beginning of the weld. As a result, the wetting behaviour is decreased and the viscosity is higher. The local build-up rate in the *z*-direction is therefore high. During welding, the base layer gets hotter. Simultaneously, the heat accumulation at the end is the highest. Since the welding power is constant, the weld acts on a continuously increasing

base-material temperature level. The wetting behaviour therefore increases and the viscosity decreases. The local build-up rate in the upwards direction is therefore lower. The real samples therefore look different depending on the welding path direction, especially at the beginning, or rather, the end (see Figure 9).

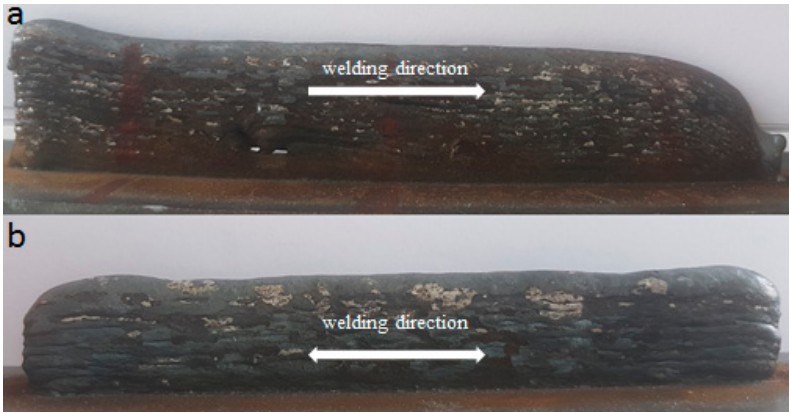

**Figure 9.** Comparison of geometry depending on the welding orientation, (**a**) discontinuous and (**b**) continuous, with $v_{wire}$ = 2.5 m/min and $v_{welding}$ = 40 cm/min for G4Si1.

- **AZ31 Results**

In addition to the walls, multi-layered tubes were also produced and numerically simulated using WAAM technology. In Figure 10, the temperature profiles are compared between experiment and numerical simulation, based on the example of AZ31. However, the simulation was carried out with only 10 layers, whereas the experiment had 20 layers.

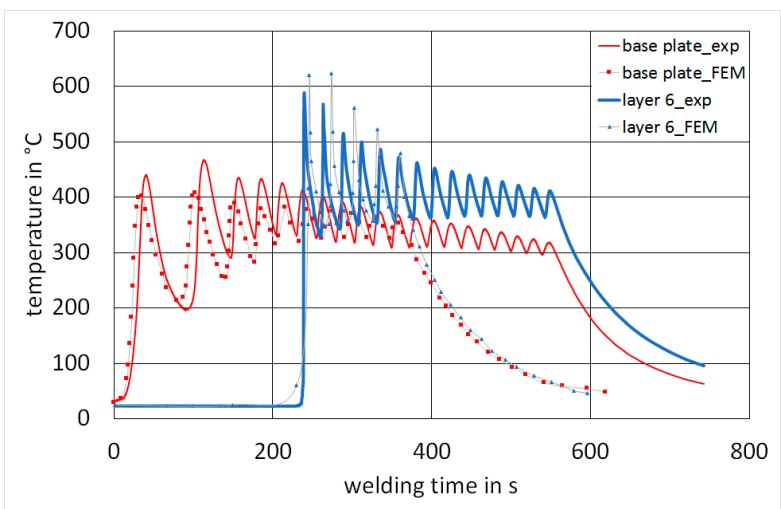

**Figure 10.** Comparison of measured (solid line) and simulated (dotted line) temperature for AZ31 in the base plate and layer 6 with $v_{wire}$ = 5.0 m/min and $v_{welding}$ = 40 cm/min.

On the one hand, there is a minimal time difference between the experimental and numerical results. This partly results from the inconsistent time steps of the recording device, which did not achieve the set measuring rate of 0.025 s. However, in the simulation, the boundary conditions were ideal and constant. The small deviations in the temperature level, as well as in the reaction time, resulted mainly from the inertia of the thermocouples. Nevertheless, summarising all temperature curves, a high accuracy and consistency between them can be seen.

## 5. Conclusions

The experimental and numerical investigations on walls and tubes of G4Si1 and AZ31, using CMT for WAAM, have shown that the welding orientation has a critical influence on the temperature field as well as on the component geometry. The aim was the experimental, but especially the numerical, investigations of welding parameters and their influence on temperature development. For realistic numerical simulation, all temperature-dependent material data were first determined without database material but with samples, which have comparable microstructure to the WAAM process. On the basis of these thermal material behaviours followed modelling, and then final implementation in the MSC Marc software. Based on this, it was possible to simulate the temperature profile in the semi-finished products very well, when the seam geometry and heat source according to Goldak were defined. Purely for convenience, the welding table was not modelled, but its heat-transfer coefficient was taken into account. The simulated temperatures in the components can be the basis for the prediction of distortion, residual stresses and, if necessary, coupling to microstructural phenomena (e.g., multiple phase transformation according to $t_{8-5}$-time for steel grades). These effects are being analysed and described in ongoing research work.

**Author Contributions:** M.G., A.H. and K.H. designed and realized the experiments; A.H. and K.H. identified and modeled the material properties; M.G. made the numerical simulation of WAAM processes; M.G. wrote the paper; B.A. and P.M. are the project coordinators and supervisors.

**Funding:** This research was funded by German Academic Exchange Service (DAAD) grant number 57347629.

**Acknowledgments:** These results are part of the subproject "Digital Engineering in Different Cultures" from the German Academic Exchange Service (DAAD) project "Higher Education Dialogue with People from Islamic Countries". The authors thank DAAD for the financial support. The authors thank also the colleagues from the Institute of Metal Forming and Professorship of Technical Thermodynamics of TU Bergakademie Freiberg for their support with material characterization.

**Conflicts of Interest:** The authors declare no conflict of interest.

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
