# Peer review of "Thermo-Mechanical Modelling of Wire-Arc Additive Manufacturing (WAAM) of Semi-Finished Products"

_metals, doi:10.3390/met8121009_

Round 1

Reviewer 1 Report

There is information that do not necessary needs to be in an abstract, line 20 for example, capabilities of the process mentioned without addressing them in the study, this however could go in the introduction. Please acknowledge irrelevant information only interrupt the reading flow.

Additionally, nowhere in the introduction have the authors mentioned the aim of the study unfortunately. They do say the current aim of numerical simulation for processes, mentioning techniques for determination of temperature fields and residual stresses. They do mention the current focus of WAAM process too, citing recent literature with no description of their own. Please acknowledge this represents a serious flaw as there is no motivation to read their work unfortunately.

Also, seems the introduction is a bit too wordy. As an example, the words manufacturing processes have been used 6 times in the first 7 sentences!

The authors addressed the material characterisation as a description of the potential causes involved in the final shape of their products. After mentioning the problems faced when collecting the data, claiming that these have minimum detrimental influence; positioning of thermocouples and peak temperatures. Although it doesn’t represent a serious flaw, feels the material characterisation falls short for a metals journal.

Worth to mention the simulation reports a fair description to the experimental data, another suggestion this may suit best to a numerical simulation journal. With no appropriate introduction however, it is difficult to know what the contribution of this paper is.

Author Response

Thank you for your helpful comments.

All was corrected and is highlighted in the word document under consideration of the remarks of a second reviewer.

Reviewer 2 Report

The paper addresses a hot topic in research nowadays, concretely it is focused on the FEM modelling of the WAAM process, one of the most promising additive manufacturing techniques for producing metallic parts at the moment; results have been validated by experimental results.

However, the paper should include additional information to make the research ready for publication in this journal, as described below.

GENERAL COMMENTS

The main interest of this paper is mostly to explain in detail how the finite element modelling has been developed, more than analysing the effect of WAAM parameters itself. Too many variables have been considered, which is interesting, but makes the paper sometimes difficult to follow as the analysis of the variables does not follow a structured methodology and there is not a design of experiments (DoE). Therefore, an effort in this direction is required.

TITLE

Please, modify the title in order to describe more clearly the content of the paper, and it can be easily distinguished from previous works of the group, as reference [8]. For example, indicate that the paper is focused on modelling of thermal behaviour.

ABSTRACT

The abstract is very generic and include general information that belongs mostly to a “introduction” section.

Please, include the specific contributions presented in this paper: variables analysed, etc,…

INTRODUCTION

Please, for classification of the additive manufacturing processes, refer also to categories included in the ISO standard:

ISO 17296-2:2015, “Additive Manufacturing. General principles. Part 2: Overview of process categories and feedstock”, International Organization for Standardization, Ginebra, 2015.

Please, include a more exhaustive state of the art, including recent works about research in WAAM techniques and finite element modelling of it.

2. FE MODEL

2.1. Material data

Please, expand this section explaining more in detail how the material data have been collected.

Please, include additional information about the software JMatPro software and include a reference of the manual in the list of references.

Please, include tables with the material data instead of referring to previous works; for example, include the Young’s modules and the flow curves.

Please, explain the “element birth and death” method for readers that are not familiar with this topic.

2.2. Boundary conditions

Please, include a figure with the initial mesh of the two main geometries (wall and pipe), in order to show details about the areas of finer meshes.

3. EXPERIMENTAL SETUP

Lines 140-144: please, refer to the sketches of figure 4 using a), b) and c).

4. SIMULATION AND EXPERIMENTS FOR WAAM.

This section contains a lot of information but it is not clearly presented. Please, make an effort to clarify which variables are analysed and why. For example, two alloys have been analysed, but results are mixed up.

Figures 6 and 7: please, indicate the material and include additional graphs for all the alloys analysed.

Lines 160-161: please, indicate which is the selection criterion for choosing the selected samples.

REFERENCES

The list of references is not exhaustive and does not include a complete list of important works recently published in WAAM field.

Please, include the name of all the authors in references, instead of using “et al.”.

Author Response

Thank you for your helpful comments.

All was corrected and is highlighted in the word document under consideration of the remarks of a second reviewer.

Only the flow curves and Young's modules from previous works is quoted because it is not new.

Round 2

Reviewer 1 Report

After including comments from the reviewing process, the manuscript shows a clear improvement. Happy to suggest the manuscript for publication.  

Reviewer 2 Report

The paper must be still rewritten according to the comments; especially more detailed information about the modelling of the problem and the methodology must be included before the paper is ready for publication.

ABSTRACT

The abstract still includes general information that belongs mostly to a “introduction” section. Please, remove the following text from the abstract:

Above all, it was necessary to produce preforms for subsequent forming processes (such as flow forming or bulk forming). The advantage of a fast AM process is that almost any material can be deposited and additional forming processes can eliminate any process-related defects (such as pores) in the material. Furthermore, the forming process can also modify the microstructure through the transformation of a dendritic cast microstructure into a globulitic microstructure, which improves mechanical properties.

INTRODUCTION

The ISO standard suggested is not well reference in the text. Please, include the right reference (EN-ISO 17296-2:2015) and include it in the list of references at the end of the paper:

ISO 17296-2:2015, “Additive Manufacturing. General principles. Part 2: Overview of process categories and feedstock”, International Organization for Standardization, Ginebra, 2015.

2. FE MODEL

2.1. Material data

This section still does not provide enough information about material modelling. It is highly recommended to include flow curves and Young's modules from previous works and to cite in the text the source (please, ask for authorization if necessary).

2.2. Boundary conditions

Please, explain the difference of number of layers between figure 3a and 3b.

4. SIMULATION AND EXPERIMENTS FOR WAAM.

This section contains a lot of information, but it is not clearly presented. Please, make an effort to clarify which variables are analysed and why. For example, two alloys have been analysed, but results are mixed up.

REFERENCES

The list of references is still not exhaustive and does not include a complete list of important works recently published in WAAM field.

Please, do not use capital letter for the title of the paper in reference [13].

Author Response

Dear reviewer,

nearly all comments are considered. But the point of criticism can not understand. Which author is missing or which must be considered? In my opion not everyone can be regarded and in each paper are different WAAM experts cited.

With warm regards.
